# Effects of Sepsis on Immune Response, Microbiome and Oxidative Metabolism in Preterm Infants

**DOI:** 10.3390/children10030602

**Published:** 2023-03-22

**Authors:** Anna Parra-Llorca, Alejandro Pinilla-Gonzlez, Laura Torrejón-Rodríguez, Inmaculada Lara-Cantón, Julia Kuligowski, María Carmen Collado, María Gormaz, Marta Aguar, Máximo Vento, Eva Serna, María Cernada

**Affiliations:** 1Division of Neonatology, University and Polytechnic Hospital La Fe (HULAFE), 46026 Valencia, Spain; 2Neonatal Research Group, Health Research Institute La Fe (IISLAFE), 46026 Valencia, Spain; 3Department of Biotechnology, Institute of Agrochemistry and Food Technology, National Research Council (IATA-CSIC), 46980 Valencia, Spain; 4Department of Physiology, University of Valencia, 46010 Valencia, Spain

**Keywords:** sepsis, preterm, immune response, oxidative stress, microbiome

## Abstract

This is a narrative review about the mechanisms involved in bacterial sepsis in preterm infants, which is an illness with a high incidence, morbidity, and mortality. The role of the innate immune response and its relationship with oxidative stress in the pathogenesis are described as well as their potential implementation as early biomarkers. Moreover, we address the impact that all the mechanisms triggered by sepsis have on the dysbiosis and the changes on neonatal microbiota.

## 1. Introduction

Neonatal sepsis is a major cause of morbidity and mortality in preterm infants. A recent meta-analysis estimates the worldwide incidence of neonatal sepsis in 28.24 cases per 1000 live births [1]. The incidence of early-onset sepsis (EOS) increases as gestational age decreases, with up to 6 cases per 1000 infants born at <34 weeks’ gestation or 20 cases per 1000 infants born at <29 weeks’ gestation. Moreover, preterm infants suffer high rates of EOS-attributable mortality (as high as 40% among extremely low gestation infants). If we take birth weight into account, the incidence of EOS in VLBW varies from 9 to 11 cases per 1000 infants [2,3].

These rates increase in case of late-onset sepsis (LOS), where are estimated to be around 12–28% among 22 to 26 weeks’ preterm infants, with a mortality close to 35% in the most vulnerable, lowest-gestation infants [4,5].

Remarkably, signs and symptoms of neonatal sepsis are subtle and nonspecific and frequently undistinguishable from clinical characteristics of preterm infants during postnatal adaptation, rendering clinical diagnosis extremely difficult. Moreover, consensus definition is still lacking [6,7].

The gold standard for the diagnosis of sepsis is a positive blood culture with an identified microorganism. However, despite the use of innovative laboratory methods, results are often delayed more than 24 h or do not yield conclusive results for the clinician. Various circumstances such as low bacteriemia, small blood volume inoculation or antibiotics administered to the mother often hamper achieving prompt and reliable results [8].

Recently, the use of multiplex polymerase chain reaction and molecular assays based on microbial genome hybridization or amplification have been tested in an attempt to detect the presence of bacteria early in the course of disease [9].

However, these techniques need a period of incubation that ranges from 4 to 6 h and a high blood volume to improve sensitivity. Moreover, bacterial PCR assays do not discriminate between viable and nonviable organisms or free and cell-associated DNA, and they do not provide enough information about antibiotic resistance [10]. For these reasons, some authors prefer to use the traditional blood culture better than these alternative methods [11]. 

The availability of a rapid and reliable diagnostic tool for neonatal sepsis still represents a major challenge for clinicians. It would allow not only early identification of infected infants and improve outcomes but also reduced exposure to unnecessary antimicrobials, subsequently avoiding the risk of inducing antibiotic resistance [12]. Furthermore, antibiotics indiscriminately affect all the commensal gut microbial ecosystem and can, therefore, alter its composition [13]. Finally, the use of antibiotics in the absence of sepsis has been related to an increased risk of mortality or major morbidity such as persistent periventricular echogenicity or echolucency on neuroimaging, chronic lung disease, and stage 3 or higher retinopathy of prematurity [14].

Since sepsis is a widespread process that is not confined to any specific organ, most studies have investigated the role of several mediators of the inflammatory cascade in the early diagnosis of sepsis. Some diagnostic biomarkers based on the immune response such as acute phase reactant proteins, procalcitonin, cytokines or cell surface antigens have been proposed to early detect bacterial sepsis [15,16] and/or to monitor response to therapy [17]. However, no single biomarker so far has shown enough diagnostic power to rule out sepsis at the time of clinical suspicion so far and, therefore, the concurrent use of several biomarkers in a sequential manner and based on immune response to achieve higher sensitivity and specificity has been recommended [18]. These immune and inflammatory responses triggered by the sepsis release proteases, cytotoxic enzymes, reactive oxygen and nitrogen species that could lead to oxidative stress and tissular damage. The pro-oxidant environment linked to the mediators of inflammation is capable of producing dysbiosis and changes in the gut microbiome (Figure 1). A global approach based on the knowledge of immune response and secondary induced changes in metabolomic and microbiome profiles should go in depth for a better management of this issue.

Considering the findings in the latest publications, the diagnosis of neonatal sepsis should focus on the host’s response and not so much on the search for the causative germ. The differences identified in the immune response, depending on the type of infectious agent, will guide the treatment, not only earlier, but also with a shorter duration. On the other hand, metabolomics will allow us to go a step further toward the investigation of profiles suggestive of sepsis in non-invasively accessible biological fluids, such as urine or saliva [19].

Sepsis activates the immune and inflammatory responses that generate oxidative stress, and both generate a pro-oxidant environment in the intestine that alters the microbiota. Not only sepsis, but also the use of antibiotic therapy as a treatment, plays a relevant role in this complex mechanism. Therefore, the main purpose of this review is to delve into the pathophysiological mechanisms of the immune response of the premature newborn against sepsis, its influence on the oxidative balance and how both factors influence the host microbiota. The relationship between these components will allow us to improve the understanding of this pathology in order to design future diagnostic, prognostic and therapeutic strategies.

## 2. Immune Response 

The immune system in early life goes through rapid and radical changes. Given the malleability of the immune system in the newborn period, interventions aimed at modulating its trajectory thus have the potential to translate into considerable reductions in the infectious disease burden with short and long-lasting benefits. However, an improved understanding of the underlying molecular drivers of early life immunity is a prerequisite to optimize interventions and transform the window of early life vulnerability into one of opportunity [20].

The immune response in neonates is triggered by the innate immunity reacting to the exposure to an infectious agent. Local immune sentinel cells such as monocytes and macrophages are involved in pathogen-associated molecular patterns (PAMPs) recognition through the activation of pattern recognition receptors (PRRs), essentially Toll-like (TLRs) receptors but also other intracellular receptors that include nucleotide-binding oligomerization domain (NOD)-like receptors (NLRs) and retinoic acid-inducible protein (RIG)-like receptors (RLRs) [21]. 

The paradigms of PAMPS are lipopolysaccharide endotoxins (LPS) of the Gram-negative bacteria surface and lipoteichoic acid (LTA) and peptidoglycan of the Gram-positive bacteria wall. The immune response initiated by PRRs yields to the production of pro-inflammatory cytokines via mitogen-activated protein kinases (MAPK) and the transcription factor nuclear factor κB (NF-κB) [22]. In an attempt to amplify the innate immune response, the production of nitric oxide (NO), leukotrienes, prostaglandins, platelet activating factor (PAF), complement, pro-inflammatory cytokines and chemokines such as interleukins (IL-1ß, IL-6, IL-8, IL-12, IL-18), interferon gamma (INF-γ) or tumor necrosis factor alpha (TNF-α) is enhanced, the vascular permeability is increased, and inflammatory cells are recruited [23,24]. Cytokines are mainly produced by activated lymphocytes and macrophages and are involved in regulating inflammation through cellular proliferation and differentiation, chemotaxis and modulation immunoglobulin secretion. However, compared to adults, septic neonates’ levels of IL-1ß, TNF-α, INF-γ and IL-12 are significantly lower due to a gestational age-related decreased production of Myeloid Differentiation Factor 88 (myd88), Interferon Regulatory Factor 5 (IRF 5) and p38 [25,26]. 

The production of pro-inflammatory cytokines leads to the activation of endothelial cells and overexpression of cellular adhesion molecules that promote leukocytes recruitment and diapedesis. Although cytokine concentration levels seem to be adequate, leukocyte chemotaxis is limited in neonates and probably related to an insufficient up-regulation of complement receptor and inhibition caused by bacterial products [27]. Complement is involved in opsonization; it has chemotactic and anaphylactic activities that increase leukocyte aggregation and local vascular permeability. In neonates, opsonization through complement is limited and has a direct relationship with gestational age. In addition, the low levels of complement proteins and their depressed function make preterm infants especially susceptible to infection [28]. Moreover, polymorphonuclear leukocytes (PMN) show qualitative and quantitative deficiencies compared to adults. During infection, medullar PMN storage quickly decreases, apoptosis is delayed and the ability to activate cytotoxic functions is increased. PMN deformability is reduced, leading to aggregation in the intravascular space and decreased diapedesis. Medullar storage is also depleted, and immature and dysfunctional forms are released [29].

In an attempt to control the inflammatory response, anti-inflammatory cytokines such as IL-4, IL-10, IL-11, IL-13 and transforming growth factor beta (TGF-ß) are released to suppress the activation of macrophages and production of pro-inflammatory cytokines. 

These mediators, especially IL-10, released by lymphocytes Th_2_, lymphocytes B and macrophages in response to TNFα, block the activation of phagocytic cells and fever, modify the expression of coagulation factors, and decrease intermediate reactive species of oxygen and nitrogen and other vasoactive markers. However, an overproduction of all these biomarkers can lead to the suppression of the immune function [23].

Both the pathogen’s characteristics and the pattern of circulating cytokines determine the differentiation of T helper precursor cells (Th) toward lymphocyte Th_1_ or Th_2_. Th_1_ cells produce INFγ, IL-2 and TNF-β that promote cellular immunity and phagocytic activity, mostly in case of intracellular infections. Th_2_ cells produce IL-4, IL-5, IL-6, IL-9, IL-10 and IL-13, triggering humoral immunity and antibodies production [30].

The effectiveness of the inflammatory response is closely related to a balanced production of cytokines. Contrarily, an excessive response could be harmful to multiple organs such as the brain, kidney, lungs, cardiovascular system, liver, bowel and microcirculation, leading to a systemic inflammatory response characteristic of an overt sepsis. The instauration of a successful treatment will rapidly return biomarkers to normal levels, organ dysfunction will be solved, and the patient’s clinical status will improve. On the contrary, ineffective treatment is characterized by the persistence of elevated biomarkers, multiorgan failure, and death [31].

In addition to the initial inflammatory response, the presence of microorganisms increases the production of innate proteins with an important immunologic function that decreases the bacterial burden by increasing the bacterial permeability depending on neutrophils. These proteins are acute phase reactants such as: collectins, lactoferrin, haptoglobin, phospholipase A2, procalcitonin, C-Reactive protein or serum amyloid A. Furthermore, sepsis promotes an increase in other serum proteins with an opsonization function such as fibronectin and natural antibodies, mostly IgM, that produce circulating B lymphocytes. However, newborn plasma shows a decreased opsonization activity and poor production of these proteins, rendering this population more susceptible to infection [32]. 

Transcriptomic profiling from whole blood can reveal diagnostic and prognostic gene signatures in acute inflammation. Therefore, it provides an excellent and exhaustive overview of host immune response against infection. Whole blood is a rich source of cells involved in immune response such as leukocytes, monocytes and macrophages. Studies based on genome-wide expression profiles have been successfully harnessed for the diagnosis of sepsis and patient stratification based on the severity of septic shock in the pediatric and adult population [33,34]. Moreover, transcriptomic profiling has been able to discriminate septic from non-septic preterm infants in the neonatal period [35,36]. These studies found that the most significant biological processes expressed in septic preterm infants were related to innate immune and inflammatory responses with an overexpression of NF-κB and cytokines pathways, while the T cell receptor pathway was under-expressed. The most common genes overexpressed not only in neonates but also in children and adults are involved in these networks: CD177 antigen (CD177), Matrix metalloproteinase-8 (MMP-8), Haptoglobin (HP), Lipocalin-2, Olfactomedin 4 (OLFM-4) or Carcinoembryonic antigen-related cell adhesion molecule (CEACAM-1) [35,36,37,38]. Moreover, a sepsis score based on some of these genes conducted in adult population has been validated in neonatal patients [39]. 

Nevertheless, the innate immune function shows some changes with postnatal age. Moreover, the innate immune development at the beginning of life compared to the response at the end of life reveals similar patterns of distinct Toll-like-receptor-mediated immune responses [40]. Septic neonates, compared to septic infants, children, and adults, showed a significantly reduced expression of genes related to TLRs, Triggering Receptor Expressed on Myeloid Cells (TREM)-1, and inducible nitric oxide synthase (iNOS) signaling pathways [41,42]. Although an overexpression of MMP8, CD177 and HP has been reported in septic neonates, fold changes are lower than those observed in children and infants. 

According to these results, recent studies in VLBW with LOS based on RNA sequencing found an overexpression of genes related to innate immune response and inflammatory processes such as IFN-α/β, IFN-γ, IL-1 and IL-6 as well as pathways involved in pathogen recognition through TLR, pro-inflammatory and inhibitory cytokine signaling, immune and hematological regulation, and altered cholesterol biosynthesis metabolism [43]. 

In this scenario, numerous studies in recent years have aimed to describe the mechanisms of the immune response, and predictive models have been proposed [44]. Moreover, transcriptomic studies have found different responses in sepsis caused by Gram-positive compared to those caused by Gram-negative bacteria not only in adults but also in children and preterm infants [45]. Preterm infants with Gram-positive sepsis showed an overexpression of genes such as CD37, CSK and TEP1 that are related to cytokine production, cell survival, and metabolic and immunomodulating responses. This equilibrium between pro- and anti-inflammatory responses could explain the better clinical outcome for Gram-positive bacterial sepsis. Previous studies in adults found significantly higher levels of pro-inflammatory cytokines (IL-1beta, IL-6, and IL-18) in Gram-positive sepsis [46]. 

Moreover, studies analyzing datasets of critically ill patients reported a different signature related to signaling and recognition pathways as well as host response mediated by neutrophils and cell survival [47]. An overexpression of genes related to cellular respiration such as NADH subunits B2 and B8 and UQCRH has been revealed in Gram-negative sepsis, while Gram-positive sepsis showed an overexpression of LATS2 associated with the transition of the mitotic cell cycle [48]. The most significant discriminative genes such as SRC, TLR6, CLC2, IL1B and CD40 are related to these pathways [48]. 

Advances in our knowledge of the intrinsic mechanisms of the immune response will not only improve our ability to attain an early and reliable diagnosis of sepsis but will also contribute to the development of new therapeutic strategies [49,50]. 

Although no single biomarker is still validated to diagnose neonatal sepsis and cannot substitute the blood culture, it seems relevant to direct our efforts of knowing how the response to sepsis works to be able to use these markers as potential diagnostic tools. 

According to the available data, the diagnosis of neonatal sepsis in the preterm newborn should be based on the combined use of several of the currently described biomarkers as well as infectious risk factors. Probably the best combination for early sepsis would be the use of PCR and IL-6 or IL-8 in an attempt to cover the earliest and later phases of the immune response [51]. In the case of late-onset sepsis, the decision tree incorporating inflammatory markers as PCR, PCT and IL-6 reached a diagnostic accuracy of nearly 88% [52].

At the moment, the duration of antibiotic therapy is based on fixed recommendations depending on the type of germ. However, significant efforts are being devoted to the search for markers that guide us toward shorter treatment regimens. A recent analysis of suspected LOS in preterm infants below 32 weeks of gestational age showed that serum IL-6 and PCT levels [53] were associated with sepsis severity and mortality risk. Therefore, some authors defend the use of PCT to guide the duration of antibiotic treatment [54,55].

## 3. Oxidative Stress and Metabolic Processes

### 3.1. Principles of Oxidative Stress

Oxidative stress, as defined by H. Sies [56], is the imbalance between oxidants and antioxidants in favor of the oxidants, leading to a disruption of redox signaling and control and/or molecular damage. The incomplete reduction in oxygen leads to the formation of free radicals, which are highly reactive species with an extremely short half-life that tend to aggressively react with nearby molecules to achieve chemical stability. Thus, free radicals attack critical components of the cell such as DNA, RNA, proteins and lipids, causing an alteration in their structure and function. To counteract or modulate the action of free radicals, antioxidant defense mechanisms are present in cells. Antioxidant defenses include enzymes such as superoxide dismutases (SOD), catalase (CAT), glutathione peroxidases (GSH-Pxs) and heme-oxygenase (HO) and non-enzymatic molecules with anti-oxidant capabilities (e.g., reduced GSH, L-cysteine, vitamin A, C and E). Free radicals are involved in the pathogenesis of numerous conditions that include processes of ischemia–reperfusion, inflammation, autophagy, and/or apoptosis. ROS that are not free radicals such as hydrogen peroxide (H_2_O_2_) are involved in physiological processes and have a relevant role as signaling molecules [57].

### 3.2. Pathophysiology of Newborn Sepsis 

The initial event in the pathophysiology of sepsis is the result of release of endotoxins from pathogenic bacteria, triggering the activation of the inflammatory cascade by polymorphonuclear leukocytes, monocytes/macrophages and lymphocytes. A major feature of sepsis is the activation of leukocytes that will release proteases, cytotoxic enzymes, ROS and reactive nitrogen species (RNS) [58] and the myeloperoxidase-derived oxidant (MPO) hypochlorous acid contributing to the killing of bacteria (HOCl) [59]. The NADPH oxidase (NOX) enzymes typically catalyze the reduction in molecular oxygen to superoxide, which is the primary product of the enzymatic reaction in most cases. The NOX family primarily mediates ROS production in leukocytes, which is central to the genesis of the inflammatory response [60]. A spontaneous or SOD-catalyzed reduction in superoxide to hydrogen peroxide may occur. H_2_O_2_ is an ROS but not a free radical, and it acts as a cell signaling molecule. H_2_O_2_ in a reducing environment will lead to an oxidative eustress and redox signaling characterized by reversible L-cysteine and methionine oxidation, reversible nitrosation and the formation of short-lived oxidized lipids. However, in bacterial sepsis, there is an excess of superoxide and nitric oxide both contributing to the formation of a burst of hydroxyl radicals, hypochlorous acid and peroxynitrite that will cause oxidative stress and damage [61,62,63].

Under reducing circumstances, H_2_O_2_ will confer stability to the NF-κB/IκB (pro-inflammatory) and Nrf2/Keap1 (antioxidant signaling) complexes. However, an imbalance toward a pro-oxidant status stress will cause a dissociation of the NF-κB/IκB complex and translocation of NF-κB to the nucleus, where it will activate the pro-inflammatory cascade, leading to the expression of cytokines and chemokines and secondary damage to tissue. Simultaneously, Nrf2 will translocate to the nucleus, activating the antioxidant-responsive elements leading to the expression of antioxidant enzymes (SOD, CAT, GPx, etc.). An imbalance in favor of the pro-oxidant status will cause a pro-inflammatory status with a profound alteration of vascular reactivity and tissue damage, which characterizes the systemic inflammatory response syndrome (SIRS) (Figure 1) [64,65].

### 3.3. Oxidative Stress in Preterm Infants with Sepsis

Several key differences in the redox signaling and redox-mediated damage clearly delineate neonatal sepsis as a separate entity from adult sepsis. In fact, the mechanisms, the inflammatory response, response to treatment, and outcome of neonatal sepsis vary not only from that of adults but vary also among neonates as a function of gestational age. Underlying differences can be attributed both to the immature innate immunity and antioxidant defense system. Extremely premature infants suffer from a relative immune deficiency, as the transplacental transfer of immunoglobulins does not occur until 32 weeks of gestation. In fact, the NOX system is thought to be a first-line mechanism of innate immunity, as there is a direct negative correlation between oxidative burst product generation and gestational age. In neonates, the immature innate immunity results in alterations in the cytokine response [66] and shows a low capacity to generate reactive oxygen species, hence limiting pro-oxidative processes in neonatal sepsis to intracellular compartment of affected tissues.

Due to the poorly developed antioxidant defenses of preterm infants, they are prone to oxidative stress [57]. In addition, they are frequently exposed to high O_2_ concentrations at the same time, providing high levels of free iron, which enhance the Fenton reaction leading to the production of highly toxic radicals [57]. Proliferating cells are particularly susceptible to oxidative stress-induced apoptosis. Consequently, neonatal sepsis-related cellular dysfunction might result in devastating effects in rapidly developing tissues, such as the brain, lungs and heart. This explains not only a considerably higher incidence of long-term effects in neonatal sepsis survivors in comparison to adults but also points out the role of redox therapy in neonatal sepsis treatment [67]. In a recent study involving couples of twin preterm infants, it was found that sepsis induced low-grade inflammation and oxidative stress in the gut mucosa as well as also changes in the gut microbiota [68]. Hence, the gene expression of exfoliated intestinal cells in septic preterm infants showed the induction of inflammatory and oxidative stress pathways in the gut and a pro-oxidant profile that caused dysbiosis in the gut microbiota, which was reflected in the predominance of *Enterobacteria* and a reduction in *Bacteroides* and *Bifidobacterium* spp. in fecal samples, leading to a global reduction in beneficial anaerobic bacteria [68].

### 3.4. Biomarkers of Oxidative Stress in Neonatal Sepsis

Reports are consistent on the importance of the involvement of ROS/RNS in neonatal sepsis and its complications. In clinical studies, determinations of antioxidants and oxidative stress and inflammation biomarkers have been employed. In newborns with sepsis, plasma levels of lipid peroxidation and protein damage were higher in infants with proven and clinical sepsis versus healthy controls [69]. In another study, cord blood oxidative stress markers were found to be significantly higher in infants with neonatal sepsis [70]. During sepsis, highly reactive HOCl is produced from MPO. By-products of the reaction of hypochlorous acid with phenylalanine contained in proteins leads to the formation of 3-chlor-tyrosine, which can be detected in biofluids and tissue by mass spectrometry [71]. Another relevant by-product derived from the oxidant action of HOCl is glutathione sulfonamide (GSA) derived from the reaction of hypochlorous acid with reduced glutathione (GSH). GSH is extremely abundant in the alveolar lining fluid. Thus, GSA has been detected in the bronchoalveolar lavage fluid in newborns with ventilation-associated pneumonia [72].

To date, literature reports on the determination of oxidative stress biomarkers in preterm infants are scarce, and none of these biomarkers have entered routine use for clinical diagnosis. Intriguingly, it could be shown that the determination of oxidative stress biomarkers of damage to proteins, DNA and lipids can be conveniently determined in non-invasive urine samples of preterm infants [71,72,73,74]. Further studies to assess the usefulness of these biomarkers for clinical diagnosis are needed.

### 3.5. Antioxidant Components of Human Milk 

Human milk is considered the gold standard for infant nutrition and is of paramount importance in preventing infections and other morbidities in neonates and especially in preterm and low-birth-weight infants and providing long term health benefits [75,76,77,78]. In spite of decreasing the availability and function of a number of bioactive substances (including lactoferrin) and probiota during processing (pasteurization and freezing) of expressed breast milk, pasteurized donor human milk has a role in reducing the risk of necrotizing enterocolitis (NEC) in preterm infants [75,76,78].

Human milk constitutes an alive complex source of multifunctional components including nutrients, cells (breast, immune and stem cells), microbiota, bioactive proteins, glycans, cytokines, and antioxidant and anti-inflammatory factors) [79]. Human milk provides a unique antioxidant profile which might be beneficial for neonates exposed to infection [80]. Moreover, human milk also confers protection to preterm infants to pro-oxidant aggression [81]. It has been shown that feeding with preterm human milk is protective against hydroxyl radical aggression as compared to formula feeding, which is reflected in the elimination of significantly fewer biomarkers of oxidative damage to proteins and DNA in the urine [13,82].

Many of the beneficial effects of human milk are linked to lactoferrin, which is a glycoprotein found in human colostrum and milk. Lactoferrin is thought to stimulate the fast proliferation of the enterocytes in the nascent intestine, thereby creating a less permeable environment as gut wall leaks and gap junctions become tighter. This potentially results in fewer colonizing pathogens disseminating to the bloodstream through translocation via a leaky gut wall [76]. Lactoferrin has a broad-spectrum antimicrobial activity against bacteria, fungi, viruses, and protozoa. Under acidic conditions (e.g., as in the stomach or in the phagolysosomes of neutrophils), the proteolysis of lactoferrin yields peptides called lactoferricins, which have enhanced antimicrobial activity [83]. The availability of iron plays an important role in sepsis because iron is obligatory for proliferation of bacteria [67] and plays a key role in oxidative stress. Hence, the sequestration of iron by lactoferrin should provide an additional beneficial effect. Another mechanism of action by which lactoferrin may impact infant gut health and gut immune development and functioning is its effect on the neonatal microbiome [67,76,77,83]. The colonization of the gastrointestinal tract was found to be different between human milk- and formula-fed infants. Whereas bifid bacteria and lactobacilli predominate in infants fed mother’s milk, *coliforms*, *enterococci*, and *Bacteroides* spp. predominate in formula-fed infants [77]. The intake of lactoferrin was capable of reducing bacterial and fungal late-onset sepsis by two-thirds with no adverse effects. These findings suggest that lactoferrin provides a significant decrease in the incidence of sepsis [76] and creates an environment for the growth of beneficial bacteria within the gut, reducing colonization with pathogenic bacteria [83].

The early host–microbe interaction is a crucial component of healthy immune and metabolic programming of term and preterm infants, which is jeopardized by sepsis and resulting oxidative stress [63]. The intake of lactoferrin could be a mechanism of counterbalancing the adverse effect of oxidative stress on the gut microbiota during neonatal sepsis, which, to make matters worse, often requires prolonged antibiotic treatments again compromising the gut microbiome.

## 4. Microbiome

During gestation and after birth, the gut is exposed to a broad spectrum of organisms, mainly bacteria but also viruses, fungi and parasites that will shape the development of the immune system. During infancy, the gut is progressively colonized by a rapidly diversifying microbiota until it reaches the adult-like pattern at around 3–5 years of life [84].

Postnatal gut microbial colonization is driven by a variety of factors such as mode of delivery, environment, gestational age (prematurity), hygienic measures, and type of diet [84,85,86]. Some studies suggest that epigenetic changes induced by intrauterine influences affect early microbial colonization and intestinal development, which may alter disease susceptibility [87].

Gut microbiota provides the necessary stimuli required for an adequate developmental programming of the epithelial barrier function, gut homeostasis, angiogenesis, and innate and adaptive immune functions [88]. It is well-recognized that disturbances in the microbiota colonization process may result in altered immune response and the risk of developing diseases or health disorders [89,90,91].

Of note, microbial dysbiosis and/or microbial infections have been associated with preterm delivery [92,93]. It has been seen that preterm neonates experience different levels of organ immaturity and an abnormal gut microbiota establishment as compared to full-term neonates [13,93,94]. In addition, neonates are also exposed to factors and interventions such as antibiotic therapy, prolonged rupture of membranes, hospital environment, C-section delivery, lack of breast-feeding and/or delayed enteral feeding that hamper the establishment of a healthy gut microbiota, thus increasing the risk of short- and long-term health problems [13,93].

A body of evidence suggests that microbial evolution begins prior to birth [95]. In preterm infants, the analysis of meconium, which reflects in utero development, has been shown not to be sterile [96,97]. Birth mode, which is an influencing factor in gut microbiota development in term infants, was not significantly associated with changes in microbial diversity, composition, specific taxa or overall microbial development in preterm infants. Maybe, this finding reflects the overall and striking influence of the NICU environment and antibiotic exposure in this population [98].

A higher presence of specific bacteria such as *Enterobacter*, *Enterococcus*, *Lactobacillus*, *Photorhabdus*, and *Tannerella* was associated to lower gestational age and then linked to preterm birth [96]. After birth, the preterm gut microbiome is characterized by higher levels of *Enterobacteriaceae*, *Enterococcaceae* and *Staphylococcaceae* and a lower presence of *Bifidobacterium* and *Bactericides* [99,100,101].

It has been also reported that the gut microbiome of extremely low birth weight (ELBW) preterm neonates harbors a lower bacterial diversity and higher presence of the Enterobacteriaceae family and the genus belonging to *Staphylococcus* and *Enterococcus* spp. [102]. Furthermore, immature intestinal response together with a higher presence of pro-inflammatory bacteria as species from *Enterobacteriaceae* family may trigger a pro-inflammatory response and favor an increased predisposition to inflammation status [103]. It has been proposed that a higher presence of Proteobacteria in the preterm gut might cause an exacerbated immune response, which may disrupt the intestinal homeostasis, intestinal permeability and alter the intestinal barrier [103]. 

In preterm infants, increased intestinal permeability favors bacterial translocation and has an effect on systemic inflammatory host response, subsequently increasing the risk of sepsis or NEC followed by the multiple organ dysfunction syndrome (MODS) [104,105]. In preterm infants, the microbiota is not only related to the development of NEC and inflammation, but seemingly, it may have a direct effect on brain function and development [106].

Mounting evidence suggests that communication between indigenous microbes, leukocytes and neurons begins during early life. Pronovost and Hsiao review evidence suggesting that interactions between the early life microbiome and the immune system are important for neurodevelopment and that alterations in microbiome–neuroimmune communication may predispose individuals to neurological and neurodevelopmental diseases [107].

Microbiota perturbations, including the elimination, reduction, or alteration of endogenous microbes, lead to altered immune development in multiple tissue sites. In the brain, the absence of the microbiome in germ-free mice alters morphological and transcriptional features of brain-resident microglia. Several recent studies reveal that manipulating the gut microbiota alters the development of brain-resident microglia of the neuroimmune system. To determine whether the microbiome impacts microglial development, transcriptomes were profiled in newborn and adult microglia from germ-free (GF) mice compared to conventionally colonized controls. Across two independent studies, adult microglia from GF mice exhibited decreased expression of genes relevant to inflammation and immune defense, such as CST7, P2RY13, MCM5 and TPI1 [108].

Neurodevelopmental disorders, such as autism spectrum disorder and schizophrenia, are co-morbid with immune dysregulation across the gut, periphery and brain, and they are associated with many immune-related genetic and environmental risk factors [109].

Both early life immune activation and microbial dysbiosis are regarded as susceptibility factors for the etiopathogenesis of neurobehavioral issues that underlie the diagnostic features of the disorders [109].

Consistent with the ability of the gut microbiota to regulate immune homeostasis, differences in the composition of the gut microbiome were important for determining the severity of inflammatory responses to immune activation and the downstream consequences on fetal brain development and adult behavior. Notably, manipulation of the offspring microbiome at weaning was sufficient to correct only a subset of behavioral abnormalities in the model [110].

In a mouse model for Alzheimer’s disease [111], early life exposure to antibiotics for one week led to persistent changes in microbiota composition with increased *Lachnospiraceae* and decreased S24–7 later in life and decreases in IL-2, IL-3, and stem cell factor (SCF) in the cerebrospinal fluid. Interestingly, early life antibiotic exposure and alterations in the gut microbiome reduced adult amyloid-β plaque burden in the cortex and hippocampus, which coincided with fewer plaque-associated, ramified microglia and astroglia, and a greater abundance of Tregs [107]. In addition, several animal models of prenatal and early postnatal stress have consistently revealed stress-induced alterations in the microbiome, with reductions in *Lactobacillus* spp. in particular, that correlate with stress-induced immunological and behavioral abnormalities relevant to anxiety and depression [112]. The transplantation of vaginal microbiomes from maternally stressed animals into C-sectioned offspring from control mothers sufficiently induced alterations in hypothalamic gene expression [113]. In another study, supplementing stressed animals with *Lactobacillus reuteri* improved behavioral impairments in the forced swim task, which was used to measure features of depression-related “despair” [63].

These studies suggest that the microbiome can tune neuroimmune interactions and neurodevelopment to alter predisposition to later-life symptoms of neurological disease.

### 4.1. Microbiome and Sepsis 

The intestinal microbiome integrates environmental inputs with genetic and immune signals that affect the host’s metabolism, immunity and response to infection. The innate immune system, located in the gut, has the ability to sense microorganisms and translate the signals into host physiological responses and the regulation of microbial ecology [114].

Sepsis induces a change in the intestinal microbiome not only due to the action of the pathogen bacteria, oxidative stress and inflammatory response but also due to the effect of the antibiotherapy. Moreover, Cernada et al. [68] reported an activation of the inflammatory and oxidative stress pathways in parallel with an altered gut microbiota, enriched with *Enterobacteriaceae,* and a reduction in *Bacteroides* and *Bifidobacterium* spp., in the gut of septic preterm infants when compared to their control twin. 

On the other hand, this dysbiosis is considered not only as a consequence but also as a possible risk factor for the appearance of sepsis and other pathologies such as necrotizing enterocolitis. Dysbiosis, understood as a change in the qualitative or quantitative microbiota composition, in preterm infants has been associated with sepsis without an increased prevalence of potential intestinal pathogens [103]. Nevertheless, a higher abundance of *Enterobacter* (belonging to *Proteobacteria* phylum) and *Staphylococcus* spp. (belonging to *Firmicutes phylum)* in preterm gut has been associated with NEC and LOS [115]. 

Moreover, lower bacterial diversity in preterm meconium is related to a higher risk of sepsis [116]. Microbial dysbiosis preceding NEC in preterm infants is characterized by increased relative abundances of *Proteobacteria* and decreased relative abundances of *Firmicutes* and *Bacteroidetes* [117]. Overall, α-diversity (Chao1) was significantly lower in the preterm microbiome at two weeks but not one week before or at the time of diagnosis of LOS [118]. This study also reported changes in microbial structure before LOS, although those changes were dependent on timing and site.

Since the use of antibiotherapy for the treatment of sepsis can affect the gut microbiome and modulate microbial communities, impacting on the abundance and diversity [119] with the side effects that this entails, further investigations are studying the possibility of restoring the microbiome through the use of probiotics, but still there is not strong evidence to put it into clinical practice [120]. 

Nowadays, there is still not evidence that the use of probiotics in the NICUS decreases the incidence of sepsis or NEC, despite have been realized trials with different probiotic strains and different quantities of probiotics [121,122]. On the other hand, clinical safety trials are lacking. A limiting factor for further studies would be the need for the microbiologist to be able to detect probiotics in blood in the setting of a neonate with sepsis who has been administered probiotics [123].

### 4.2. Bacterial Metabolism

Metabolomics globally evaluates the totality of the endogenous metabolites in patient’s body, at the same time reflecting the gene function, enzyme activity and degree of organ dysfunction in sepsis. 

It is known that bacterial metabolism in the gut is expected to contribute to the abundances of metabolites detected in fecal samples. Wandro et al. found that *Staphylococcus* had the most positive correlations, including several classes of sugar metabolites, organic acids, and central metabolites. Fatty acids, lipid metabolism, and amino acids positively correlated with the commonly abundant gut colonizers *Enterobacteriaceae* and *Bacteroides* and negatively with the common low-abundance colonizers *Sthaphylococcus* and *Enterococcus. Bacteroidetes* were found to positively correlate with succinate (*r* = 0.85) [124].

Nevertheless, metabolic dysbiosis is not mandatorily accompanied by appreciable quantitative and/or qualitative changes in microbiota composition called taxonomic dysbiosis, indicating the need for further research in this area to find different approaches to its assessment using metabolomics (metabolic fingerprinting, metabolic profiling, meta-metabolomics). This line of research is currently in full swing, seeing that metabolites concentrations in colon, blood, urine or exhaled air, as well as the metabolic profiles of examined substrates, can serve as biomarkers [125].

## 5. Conclusions 

The sepsis in preterm infants triggers the innate immune and inflammatory responses, leading to oxidative stress and tissular damage. The pro-oxidant environment linked to the mediators of inflammation is capable of producing dysbiosis and changing the gut microbiome. The knowledge of all these mechanisms and the study of transcriptomic, metabolomic and microbiome profiles could help us reach an earlier and more accurate diagnosis. We conclude that this is a field still to be explored, where more research should be carried out to be able to transfer it to the clinical practice in order to achieve a reduction in the morbidity and mortality of this population.

## Figures and Tables

**Figure 1 children-10-00602-f001:**
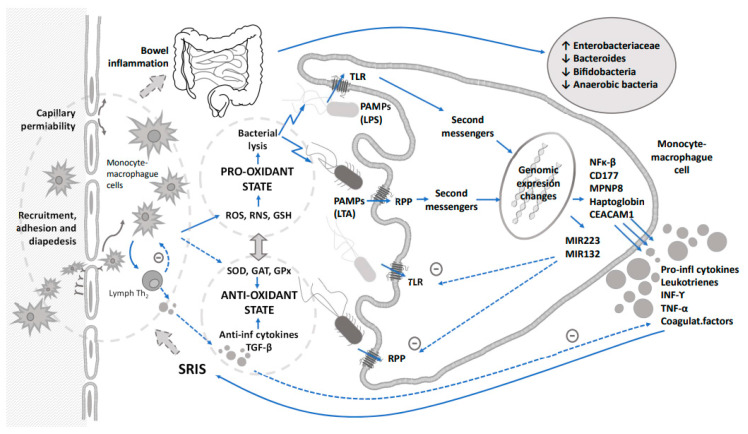
Integrative relationship between immune response triggered by sepsis, oxidative stress and their impact on dysbiosis. Dashed arrows indicate counterregulation of inflammation and oxidative stress.

## Data Availability

No new data were created.

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
