# Peer review of "Effects of Sepsis on Immune Response, Microbiome and Oxidative Metabolism in Preterm Infants"

_children, 2023, doi:10.3390/children10030602_

Round 1

Reviewer 1 Report

1. This Review involve  EFFECTS OF SEPSIS ON IMMUNE RESPONSE,  MICROBIOME AND OXIDATIVE METABOLISM in  in preterm infants; not reflected in title.

2. The incidence of early-onset sepsis (EOS) ranges from 1.5 to 1.9%, while late-onset sepsis (LOS) rates are estimated to be around 20% in very low birth weight (VLBW) infants with an overall mortality close to 20% [2-3].

You have provided the data of study 21 years back, no more relevant now-a day. Moreover, you clear about your target population. Preterm infant and VLBW infants are not same. In abstract you are telling about preterm infant and provided data of VLBW babies on introduction.

3. role of microbiota in the development and prevention of sepsis and NE is the conclusion, the title is not matching to your findings.

4. The manuscript looking bulky. Many things are collected and summed up. No clear intention is visible. What is the knowledge gap authors try to filling is not defined properly.

5. Authors should suggest the biomarkers need to be evaluated at what point of time of preterm neonatal sepsis, so that early diagnosis could be made and bet possible treatment could be initiated.

6. Also suggest the future diagnostic markers on various phages of study, need to be evaluated/ further research is needed. 

7. What are the biomolecules with highest sensitivity/specificity to be investigated that aid on treatment process. As till date most of the centers using clinical evidence and few lab. Parameters to treat and prognosticate the LBW babies. Is there any prognostic markers ?

Author Response

Thank you for your comments and suggestions. We have taken all into account in an attempt to improve the quality of the manuscript.

We are providing our answers to your questions.

  1. This Review involve  EFFECTS OF SEPSIS ON IMMUNE RESPONSE,  MICROBIOME AND OXIDATIVE METABOLISM in  in preterm infants; not reflected in title.

We appreciate the comment, to clarify a possible confusion, we also consider that it would be convenient to modify the title as follows:

"Effects of sepsis on immune response, microbiome and oxidative metabolism in preterm infants".

  1. The incidence of early-onset sepsis (EOS) ranges from 1.5 to 1.9%, while late-onset sepsis (LOS) rates are estimated to be around 20% in very low birth weight (VLBW) infants with an overall mortality close to 20% [2-3].

You have provided the data of study 21 years back, no more relevant now-a day. Moreover, you clear about your target population. Preterm infant and VLBW infants are not same. In abstract you are telling about preterm infant and provided data of VLBW babies on introduction.

We have tried to provide our own data, but reviewer is true that they are out of step. We have included more recent data and differentiate data from preterm and very low birth weight infants, who are the most common affected population. We have changed the text, that now reads as follows:

“Neonatal sepsis is a major cause of morbidity and mortality in preterm infants. A recent meta-analysis estimates the worldwide incidence of neonatal sepsis in 28,24 cases per 1000 live births. The incidence of early-onset sepsis (EOS) increases as gestational age decreases, with up to 6 cases per 1000 infants born at <34 weeks' gestation or 20 cases per 1000 infants born at <29 weeks' gestation. Moreover, preterm infants suffer high rates of EOS-attributable mortality (as high as 40% among extremely low gestation infants). If we take birth weight into account, the incidence of EOS in VLBW varies from 9 to 11 cases per 1000 infants”.

References:

  1. Fleischmann C, Reichert F, Cassini A, Horner R, Harder T, Markwart R, et al. Global incidence and mortality of neonatal sepsis: a systematic review and meta-analysis. Arch Dis Child. 2021;106:745–52.
  2. Puopolo KM, Benitz WE, Zaoutis TE, COMMITTEE ON FETUS AND NEWBORN, COMMITTEE ON INFECTIOUS DISEASES. Management of Neonates Born at ≤34 6/7 Weeks’ Gestation With Suspected or Proven Early-Onset Bacterial Sepsis. Pediatrics. 2018;142:e20182896.
  3. Flannery DD, Puopolo KM. Neonatal Early-Onset Sepsis. Neoreviews. 2022;23:756–70

“These rates increase in case of late-onset sepsis (LOS), where are estimated to be around 12-28% among 22 to 26 weeks’ preterm infants, with a mortality close to 35% in the most vulnerable, lowest-gestation infants”.

References:

  1. Brumbaugh JE, Bell EF, Do BT, Greenberg RG, Stoll BJ, DeMauro SB, et al. Incidence of and Neurodevelopmental Outcomes After Late-Onset Meningitis Among Children Born Extremely Preterm. JAMA Netw Open. 2022;5:e2245826.
  1. Coggins SA, Glaser K. Updates in Late-Onset Sepsis: Risk Assessment, Therapy, and Outcomes. Neoreviews. 2022;23:738–55.
  1. Role of microbiota in the development and prevention of sepsis and NE is the conclusion, the title is not matching to your findings.

We agree with the reviewer that the NEC is not a part of our findings since our research is focused on sepsis. We have reconsidered it and have eliminate the sepsis and NEC section, and also changed the conclusions, that now reads as follows:

“The sepsis in preterm infants triggers the innate immune and inflammatory responses, leading to oxidative stress and tissular damage. The prooxidant environment linked to the mediators of inflammation are capable of producing dysbiosis and changing the gut microbiome. The knowledge of all these mechanisms and the study of transcriptomic, metabolomic and microbiome profiles, could help us to reach an earlier and more accurate diagnosis. We conclude that this is a field still to be explored, where more research should be carried out to be able to transfer it to the clinical practice in order to achieve a reduction in the morbidity and mortality of this population”.

  1. The manuscript looking bulky. Many things are collected and summed up. No clear intention is visible. What is the knowledge gap authors try to filling is not defined properly.

Following the reviewer comment, we have included a sentence in the introduction section to explain the meaning and the common thread of the manuscript. Moreover we have changed the section’s order and the abstract to better understand.

“These immune and inflammatory responses triggered by the sepsis release proteases, cytotoxic enzymes, reactive oxygen and nitrogen species that could lead to oxidative stress and tissular damage. The prooxidant environment linked to the mediators of inflammation is capable of producing dysbiosis and changes in the gut microbiome. A global approach based on the knowledge of immune response and secondary induced changes in metabolomic and microbiome profiles should be go in depth to a better management of this issue”.

“Abstract: This is a narrative review about the mechanisms involved in bacterial sepsis in preterm infants, an illness with a high incidence, morbidity, and mortality. The role of the innate immune response and their relationship with oxidative stress in the pathogenesis are described, as well as their potential implementation as early biomarkers. Moreover, we address the impact that all the mechanisms triggered by sepsis have on the dysbiosis and the changes on neonatal microbiota”.

  1. Authors should suggest the biomarkers need to be evaluated at what point of time of preterm neonatal sepsis, so that early diagnosis could be made and bet possible treatment could be initiated.

According to the available data, the diagnosis of neonatal sepsis in the preterm newborn should be based on the combined use of several of the currently described biomarkers as well as infectious risk factors. Probably the best combination for early sepsis would be the use of PCR and IL-6 or IL-8 in an attempt to cover the earliest and later phases of the immune response. In the case of late-onset sepsis, the decision tree incorporating inflammatory markers as PCR, PCT and IL-6 reached a diagnostic accuracy of nearly 88%.

Eichberger J, Resch E, Resch B. Diagnosis of Neonatal Sepsis: The Role of Inflammatory Markers. Front Pediatr. 2022 Mar 8;10:840288. doi: 10.3389/fped.2022.840288. PMID: 35345614; PMCID: PMC8957220.

Cao I, Lippmann N, Thome UH. The Value of Perinatal Factors, Blood Biomarkers and Microbiological Colonization Screening in Predicting Neonatal Sepsis. J Clin Med. 2022 Oct 1;11(19):5837. doi: 10.3390/jcm11195837. PMID: 36233706; PMCID: PMC9571877.

  1. Also suggest the future diagnostic markers on various phages of study, need to be evaluated/ further research is needed. 

We have moved the reference to future diagnostic biomarkers to immune response section. Although no single biomarker is still validated to diagnose neonatal sepsis and cannot substitute the blood culture, we tried to focus on the importance of knowing how the response to sepsis works to be able to use these markers as potential diagnostic tools.

  1. What are the biomolecules with highest sensitivity/specificity to be investigated that aid on treatment process. As till date most of the centers using clinical evidence and few lab. Parameters to treat and prognosticate the LBW babies. Is there any prognostic markers ?

We have tried to answer this interesting question by adding the following paragraph:

“At the moment, the duration of antibiotic therapy is based on fixed recommendations depending on the type of germ. However, significant efforts are being devoted to the search for markers that guide us towards shorter treatment regimens. A recent analysis of suspected LOS in preterm infants below 32 weeks of gestational age showed that serum IL-6 and PCT levels, were associated with sepsis severity and mortality risk. Therefore, some authors defend the use of PCT to guide the duration of antibiotic treatment”.

References:

  1. Li T, Li X, Liu X, Zhu Z, Zhang M, Xu Z, Wei Y, Feng Y, Qiao X, Yang J, Dong G. Association of Procalcitonin to Albumin Ratio with the Presence and Severity of Sepsis in Neonates. J Inflamm Res. 2022 Apr 12;15:2313-2321. doi: 10.2147/JIR.S358067. PMID: 35437348; PMCID: PMC9013250.

  1. Mathur NB, Behera B. Blood Procalcitonin Levels and Duration of Antibiotics in Neonatal Sepsis. J Trop Pediatr. 2019 Aug 1;65(4):315-320. doi: 10.1093/tropej/fmy053. PMID: 30137640.

  1. Fugit RV, McCoury JBM, Bessesen MT. Procalcitonin for sepsis management: Implementation within an antimicrobial stewardship program. Am J Health Syst Pharm. 2022 Nov 13:zxac341. doi: 10.1093/ajhp/zxac341. Epub ahead of print. PMID: 36371732.

Reviewer 2 Report

This comprehensive review goes into great detail regarding many different processes at work in neonates in response to sepsis. It provides groundwork for others to consider an area of interest to pursue for greater study. 

Consider amending the title to include the primary population discussed. For example, "Effects of Sepsis on Immune Response, Microbiome, and Oxidative Metabolism of Preterm Infants." This will help provide context for the reader outright. 

On page 6, the sentence beginning on line 283 is a bit confusing. For one, I'm not sure "exposition" is the appropriate word to convey the intended meaning. In addition, if appropriate, consider truncating the sentence to something like "A body of evidence suggests that microbial evolution begins prior to birth." The following supporting statements discussing meconium analysis then clarify the intention.  

Figure 1, on page 10, is not referenced anywhere within the manuscript text. Please include where it should be referenced within the manuscript. 

Page 10, line 494 - the abbreviation H2O2 appears in the text however I can't find that it was previously defined as hydrogen peroxide - please include this. 

In the Conclusions section starting on page 13, please briefly comment more specifically on areas of future research to be developed based on what is known from this review - for example, sepsis biomarkers, metabolomics etc.

Consider briefly mentioning the use of probiotics either in the conclusion statements about microbiome or within the "Sepsis and NEC section."

Author Response

Thank you for your comments and suggestions. We have taken all into account in an attempt to improve the quality of the manuscript.

We are providing our answers to your questions.

This comprehensive review goes into great detail regarding many different processes at work in neonates in response to sepsis. It provides groundwork for others to consider an area of interest to pursue for greater study. 

Consider amending the title to include the primary population discussed. For example, "Effects of Sepsis on Immune Response, Microbiome, and Oxidative Metabolism of Preterm Infants." This will help provide context for the reader outright. 

We appreciate the comment, to clarify a possible confusion, we also consider that it would be convenient to modify the title as follows:

"Effects of sepsis on immune response, microbiome and oxidative metabolism in preterm infants".

On page 6, the sentence beginning on line 283 is a bit confusing. For one, I'm not sure "exposition" is the appropriate word to convey the intended meaning. In addition, if appropriate, consider truncating the sentence to something like "A body of evidence suggests that microbial evolution begins prior to birth." The following supporting statements discussing meconium analysis then clarify the intention.  

We fully agree with the reviewer's comment. We have modified the sentence to try to make it easier to understand.

“A body of evidence suggests that microbial evolution begins prior to birth”.

Figure 1, on page 10, is not referenced anywhere within the manuscript text. Please include where it should be referenced within the manuscript. 

We corrected this error by including the reference on line 96 an 688. We have also moved the figure at the end of the manuscript in an attempt to summarize all the findings.

Page 10, line 494 - the abbreviation H2O2 appears in the text however I can't find that it was previously defined as hydrogen peroxide - please include this. 

We include the abbreviation “H2O2” on line 328.

In the Conclusions section starting on page 13, please briefly comment more specifically on areas of future research to be developed based on what is known from this review - for example, sepsis biomarkers, metabolomics etc.

In this regard, we would like to add the following paragraph on line 100:

“Considering the findings in the latest publications, the diagnosis of neonatal sepsis should focus on the host's response, and not so much on the search for the causative germ. The differences identified in the immune response, depending on the type of infectious agent, will guide the treatment, not only earlier, but also with a shorter duration. On the other hand, metabolomics will allow us to go a step further, towards the investigation of profiles suggestive of sepsis in non-invasively accessible biological fluids, such as urine or saliva”.

Ref: 19. Renwick VL, Stewart CJ. Exploring functional metabolites in preterm infants. Acta Paediatr. 2022 Jan;111(1):45-53. doi: 10.1111/apa.16146. Epub 2021 Oct 28. PMID: 34626496.

We have also change the conclusion statement:

“Conclusion: The sepsis in preterm infants triggers the innate immune and inflammatory responses, leading to oxidative stress and tissular damage. The prooxidant environment linked to the mediators of inflammation is capable of producing dysbiosis and changing the gut microbiome. The knowledge of all these mechanisms and the study of transcriptomic, metabolomic and microbiome profiles could help us to reach an earlier and more accurate diagnosis. We conclude that this is a field still to be explored, where more research should be carried out to be able to transfer it to the clinical practice in order to achieve a reduction in the morbidity and mortality of this population”.

Consider briefly mentioning the use of probiotics either in the conclusion statements about microbiome or within the "Sepsis and NEC section."

Since NEC is not a part of our research, which is focused on sepsis, we have reconsidered it and have eliminate the sepsis and NEC section.

Misprints: All misprints have been corrected

Font error on line 297 “Enterococcaceae

Font error on line 355 “Lachnospiraceae

Font error on line 366 “Lactobacillus

Font error on line 369 and 370 “Lactobacillus reuteri

Font error on line 436 “Enterobacteriaceae” and “Bacteroides”

Font error on line 438 “Bacteroidetes”

Round 2

Reviewer 1 Report

Well revised. May be accepted.